# Shortening the Standard Testing Time for Residual Biogas Potential (RBP) Tests Using Biogas Yield Models and Substrate Physicochemical Characteristics

Yanxin Liu [1], Weisi Guo [2], Philip Longhurst [1] and Ying Jiang [1,*]

1   School of Water, Energy and Environment, Cranfield University, Cranfield MK43 0AL, UK
2   School of Aerospace, Transport and Manufacturing, Cranfield University, Cranfield MK43 0AL, UK
*   Correspondence: y.jiang@cranfield.ac.uk; Tel.: +44-(0)-1234-75-4492

**Abstract:** The residual biogas potential (RBP) test is a procedure to ensure the anaerobic digestion process performance and digestate stability. Standard protocols for RBP require a significant time for sample preparation, characterisation and testing of the rig setup followed by batch experiments of a minimum of 28 days. To reduce the experimental time to obtain the RBP result, four biogas kinetic models were evaluated for their strength of fit for biogas production data from RBP tests. It was found that the pseudo-parallel first-order model and the first-order autoregressive (AR (1)) model provide a high strength of fit and can predict the RBP result with good accuracy (absolute percentage errors < 10%) using experimental biogas production data of 15 days. Multivariate regression with decision trees (DTs) was adopted in this study to predict model parameters for the AR (1) model from substrate physicochemical parameters. The mean absolute percentage error (MAPE) of the predicted AR (1) model coefficients, the constants and the RBP test results at day 28 across DTs with 20 training set samples are 4.76%, 72.04% and 52.13%, respectively. Using five additional data points to perform the leave-one-out cross-validation method, the MAPEs decreased to 4.31%, 59.29% and 45.62%. This indicates that the prediction accuracy of DTs can be further improved with a larger training dataset. A Gaussian Process Regressor was guided by the DT-predicted AR (1) model to provide probability distribution information for the biogas yield prediction.

**Keywords:** RBP test; biogas yield models; decision trees; Gaussian process; regression

## 1. Introduction

Digestate is a by-product from the anaerobic digestion (AD) process. Due to its high nutrient value, it can be used as soil improver or fertiliser if the digestate is proved to be valorised and can meet relevant quality standards [1]. The digestate stability can be evaluated with a residual biogas potential (RBP) test. The test typically is required to be carried out under mesophilic conditions for at least 28 days with an appropriate inoculum-to-substrate ratio and micro- and macronutrients supplemented to avoid the inhibition of biogas production [2]. The digestate is considered to have consistent quality if the RBP test biogas yield is below 0.25 L/g volatile solids (VS), as recommended in the Publicly Available Specification 110 (PAS110), which is a key element of the UK Government's anaerobic digestion quality protocol [3].

The 28-day continuous monitoring of biogas production in an RBP test is time-consuming and onerous for commercial AD operators. This limits the adoption of RBP tests and regulated markets for digestate. There have been many attempts to find alternative approaches to RBP that offer rapid tests result. These include assessing acid production after the inhibition of methanogenesis [4–6] and assessing the digestion of the organic fraction of the digestate after separation of the microbial cell component [7]. Nevertheless, both approaches are of great complexity and further research is needed [7].

Additionally, some researchers have attempted to relate the RBP test results to digestate physicochemical characteristics. For example, the theoretical biogas potential calculated based on the stoichiometric methane conversion from volatile fatty acid (VFA) concentrations and the soluble chemical oxygen demand (sCOD) in the digestate sample were suggested in PAS110 as preliminary pass/fail indicators for the RBP tests [8]. Although these preliminary indicators provide useful information about digestate stability, they cannot be correlated with RBP values or provide reaction kinetics that can be used to reveal further digestion performance information including inhibition. In a previous work reviewing the application of the RBP test for PAS110 [9], correlations between RBP values and various characteristics, including VFA and sCOD, were investigated. Although some low to moderate levels of correlations were found for total VFA, total solids and volatile solids, which account for 40%, 36% and 29% of variation in RBP values, respectively, none of the indicators are sufficiently reliable to predict the RBP values accurately [9].

Other researchers have evaluated using empirical biogas production models, including first-order kinetic and Gompertz models, and experimental biogas yield data from the initial stage of the RBP tests to fit specific accumulative biogas production data from RBP tests [10–12]. This has led to a promising experimental and modelling 'hybrid' approach using experimental data collected from a shorter RBP duration (3–7 days instead of 28 days) to calculate model parameters, and then predict the ultimate biogas production. However, the accuracy of prediction is not sufficient to warrant the replacement of RBP with this hybrid approach [12]; therefore, further improvement of the modelling process is required.

In this research, we evaluated the strength of fit for four biogas yield kinetic models including first-order kinetic, modified Gompertz, pseudo-parallel first-order kinetic and autoregressive (AR) time-series to describe the RBP test biogas production process. The models are then calibrated using experimental data collected from shorter RBP tests (5, 10, 15, 20 and 25 days) to calculate model parameters that are then used to predict ultimate biogas production.

In a previous work [9], although using conventional statistical methods, no significant correlations were found between key physicochemical parameters of digestate samples with RBP results. Due to the potential interplay of these parameters, which can influence the RBP results and reaction kinetics, the correlations may be deeply hidden.

Machine learning techniques including multivariate nonlinear regression analysis with decision trees (DT) were applied to predict the parameters of the biogas production model from the physicochemical characteristics of digestate samples. Compared with other multivariate nonlinear regression methods, the DT method is particularly suitable for a training dataset with limited sample size in this study [13]. The uncertainties of the predicted biogas yield were then assessed using a Gaussian process regressor (GPR).

The data processing framework described in this work can potentially have a wider application for other complex biochemical processes that are influenced by multiple physicochemical parameters of the reaction system.

## 2. Materials and Methods

### 2.1. Digestate Samples and RBP Test

The sampling point of the 25 digestate samples was the outlet of the final tank from which the biogas was collected. The AD plants involved in the study are anonymised and coded as ADP1–25. The inoculum was from the anaerobic digester at Millbrook Wastewater (WW) Treatment Plant at Southampton, UK. The RBP test followed the standard procedure described in the PAS110 [3]. Samples were tested in triplicate against two positive controls and three inoculum-only controls.

### 2.2. Analytical Methods

Twenty physicochemical characteristics were analysed for each digestate, including VFAs (total VFA and acetate), total ammoniacal nitrogen (TAN), total Kjeldahl nitrogen (TKN), alkalinity (total alkalinity (TA), partial alkalinity (PA) and intermediate alkalinity

(IA)), TS, VS, pH, COD (total COD and sCOD), trace element (TE) concentrations (cobalt (Co), iron (Fe), molybdenum (Mo) and nickel (Ni)), calorific value (CV) and elemental compositions (C, H, N). TAN/TKN and IA/PA were calculated as two extra metrics. TAN/TKN represents the relative contents of ammonia nitrogen and organically bonded nitrogen and thus how ready the substrate is for microorganism degradation [14]. IA/PA is an indicator of VFA accumulation in the AD process. Ripley et al. (1986) [15] suggest IA/PA < 0.3 indicates a stable state of the anaerobic process.

Determination of VFAs is based on the SCA (1979) [16] method Determination of Volatile Fatty Acids in Sewage Sludge. Supernatant layer from digestate centrifugation with 10% formic acid were quantified in a Shimadzu GC-2010 gas chromatograph with a flame ionisation detector and a capillary column type SGE BP-21. TAN and TKN were determined using a Kjeltech digestion block and steam distillation unit, according to the manufacturer's instructions (Foss Ltd., Warrington, UK). Alkalinity was measured by titration with 0.25 N $H_2SO_4$ to endpoints of pH 5.75 and 4.3 in order to allow calculation of TA, PA and IA [15]. TS and VS were determined with Standard Method 2540 G (APHA, 2005). Total COD and sCOD were analysed by adapting the closed reflux titrimetric method of 5220C, APHA [17]. TE concentrations were determined using ICP-MS or ICP-OES at a UKAS-accredited commercial laboratory (Severn Trent Services, Coventry, UK) after in-house hydrochloric–nitric acid digestion [18]. CV was measured with a CAL2k-ECO bomb calorimeter (CAL2k, Digital Data Systems, Gauteng, South Africa). Elemental C, H, and N analysis was performed using a Flash EA-1112 elemental analyser (Thermo Finnigan, Cheshire, UK).

### 2.3. Assessing the Strength of Fit for Biogas Production Kinetic Models

Four biogas-production kinetic models commonly used to estimate the kinetic constants of the AD process were compared for their abilities to fit the biogas production of RBP tests. These models include three empirical models (first-order kinetic, modified Gomperz and pseudo-parallel first-order models) and a time-series model (first-order autoregressive). The strength of a kinetic model to accurately fit the experimental biogas production data was measured using $R^2$ values, which indicate the percentage of the variance in the responses explained by a model.

Additionally, these models were used to predict the 28-day RBP test results by fitting the models with initial 5, 10, 15, 20 and 25 days' experimental data using the Matlab R2021b Curve Fitting Toolbox. Based on the absolute percentage error (APE) between the experimental data and model-predicted results, the accuracy of prediction and duration of experimental data required to obtain a sufficiently accurate prediction were investigated for the following four models:

(1) First-order model (FO): The FO model (Equation (1)) is derived from the assumptions that the substrate degradation is a first-order reaction with hydrolysis as the speed-limiting step and the cumulative biogas yield is proportional to the amount of substrate degraded (Equation (2)) [19,20].

$$y(t) = y_m \left( 1 - e^{-kt} \right) \tag{1}$$

$$\frac{dc}{dt} = -kc \quad \frac{c_0 - c}{c_0} = \frac{y}{y_m} \tag{2}$$

where $y(t)$ is the cumulative biogas yield at time $t$, $k$ is the first-order rate constant, $y_m$ is the maximum cumulative gas production, $c$ is the concentrate of the substrate and $c_0$ is the initial substrate concentration.

(2) Modified Gomperz model (MG): The modified Gomperz model (Equation (3)) is derived from the Gomperz model, which is used to describe the microbial activity and has a signature sigmoid shape [21,22]. It describes the biogas production in terms of the exponential growth rates and lag phase duration of anaerobic degradation microorganisms [11].

$$y(t) = y_m \times \exp\left(-\exp\left(\frac{R}{y_m} \times e \times (\lambda - t) + 1\right)\right) \tag{3}$$

where $y(t)$ is the cumulative gas production at time $t$, $y_m$ is the maximum cumulative gas production (mL $CH_4$/gVS), $R$ is the maximum gas production rate (mL $CH_4$/gVS/d) and $\lambda$ is the lag phase period or minimum time to produce biogas (days).

(3) Pseudo-parallel first-order model (PP): The pseudo-parallel first-order model (Equation (4)) is considered to be more suitable for describing the biogas yield of mixtures of substrates with different kinetic rates (rapid and slow) [23].

$$y(t) = y_m\left(1 - Pe^{-k_1 t} - (1 - P)e^{-k_2 t}\right) \tag{4}$$

where $y(t)$ is the cumulative gas production at time $t$, $y_m$ is the maximum cumulative gas production (mL $CH_4$/g VS), $P$ is the the proportion of the readily degradable material, $k_1$ is the first-order rate constant for readily degradable material, and $k_2$ is the first-order rate constant for less readily degradable material.

(4) First-order autoregressive model (AR (1)): AR (1) is a time-series model that predicts the present timestep based on the observations from previous timesteps. The autocorrelation function (ACF) (autocorrelation between timesteps) plots for all the RBP test biogas yield samples gradually trail off (Figure 1a, using ADP20 as an example). Therefore, the biogas production process is an AR process. Many time-series models that essentially model the randomness of the time series data need the trends in the data to be removed, in other words, to ensure the stationarity of the data. However, the application of the AR model does not intrinsically require transforming the data into stationary data. Thus, the biogas yield data were not converted to a stationary process in this study.

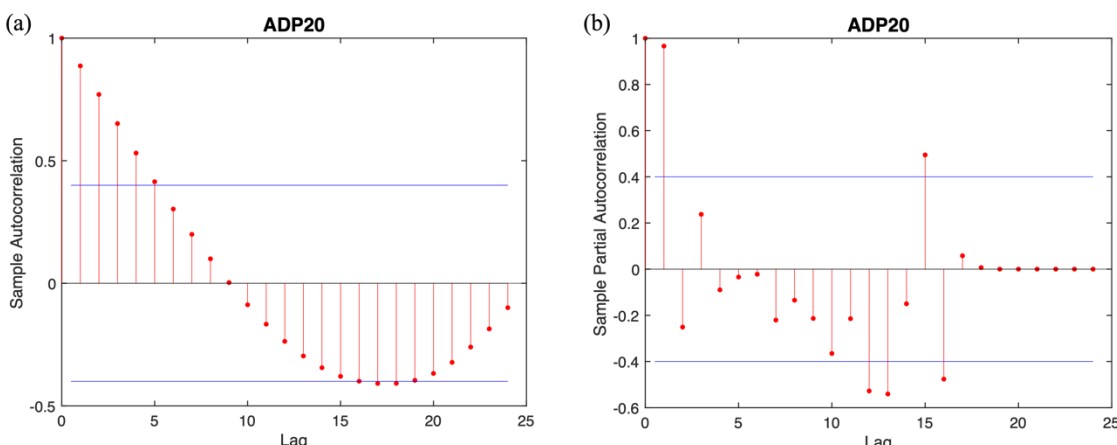

**Figure 1.** The ACF (**a**) and PACF (**b**) of sample ADP20 biogas production time series from day 4. The blue lines are the confidence bounds for a significant correlation. The red lines are the autocorrelation and partial autocorrelation between the current time step and the time steps of different lags.

For most of the RBP test data, the cumulative biogas production curves typically exhibit a rapid gas-production stage in the initial three days followed by a noticeable reduction in biogas production rate. This means one set of AR model parameters generally will result in a poor fitting for both the rapid-production stage and the later stage after the initial three days. In this study, to simplify the modelling process, the AR (1) model was only used to model biogas production from day 4 of the RBP tests. The partial autocorrelation function (PACF) (partial autocorrelation between timesteps) plots of data from day four for most of the biogas time series samples were cut off after one lag (Figure 1b, taking ADP20 as an example). Therefore, the biogas production from day four was an AR (1) process (Equation (5)).

The upper asymptote of an AR (1) process is determined by its unconditional mean $\mu = c/(1 - \beta)$, which corresponds to the ultimate biogas yield of a substrate.

$$X_t = \beta X_{t-1} + c + \varepsilon_t \tag{5}$$

where $X_t$ is the response of the present timestep, the $X_{t-1}$ is the response of the previous one timestep, $\beta$ is the coefficient, $c$ is the constant and $\varepsilon_t$ is the white noise.

### 2.4. Prediction of Biogas Kinetic Model Coefficients and Constants Using Decision Tree Multivariate Regression Method

2.4.1. Decision Tree Multivariate Regression Method

The best-performing model out of the four kinetic models was further studied using decision tree (DT) multivariate regression analysis. DT is a non-parametric supervised learning technique that can be applied for both regression analysis and classification, providing piecewise constant approximations as prediction results [24].

DTs were trained using all physicochemical characteristics listed in Tables 1 and 2 as predictors to predict the coefficients and constants of the best-performing biogas production kinetic model. First, RBP results of 20 digestate samples (out of the 25-sample dataset) were used to train the DT model with the five remaining samples as the test dataset for model validation. The training process generated 53,130 groups of different combinations of 20 samples when choosing out of the 25. For practical reasons, a subset of 5000 groups were selected randomly to evaluate the absolute percentage error (APE) and mean absolute percentage error (MAPE) (Equation (6)) of the predicted biogas production model parameters and the APE of the calculated ultimate biogas yield. Then, the DT model was trained and validated using the leave-one-out (LOO) cross-validation method, which apply four additional set of data to the training data to verify if the prediction accuracy would improve.

$$APE_i = \left| \frac{y_i - \hat{y}_i}{y_i} \right| \quad MAPE = \frac{1}{n} \sum_{i=1}^{n} \left| \frac{y_i - \hat{y}_i}{y_i} \right| \tag{6}$$

where $y_i$ is the model parameter derived by fitting the model to the entire 28-day RBP test data, $\hat{y}_i$ is the model parameter inferred from the physicochemical characteristics by DT and $n$ is the number of training set samples.

**Table 1.** RBP test results on day 28 (d-28) and the physicochemical characteristics of 25 samples (Part 1).

| ADP No. | d-28 L/g VS | Acetate | Total-VFA mg/L | TAN | TKN g N/kg | TAN/TKN | TA | PA kg/kg CaCO₃ | IA | IA/PA | TS | VS g/kg |
|---|---|---|---|---|---|---|---|---|---|---|---|---|
| 1 | 0.13 | 178.18 | 193.57 | 1.70 | 3.53 | 0.48 | 12.52 | 9.03 | 3.49 | 0.39 | 49.84 | 37.12 |
| 2 | 0.18 | 29.08 | 37.35 | 0.62 | 3.28 | 0.19 | 7.97 | 6.42 | 1.55 | 0.24 | 49.11 | 30.39 |
| 3 | 0.13 | 846.78 | 1016.41 | 7.98 | 12.37 | 0.64 | 33.38 | 25.87 | 7.51 | 0.29 | 93.28 | 67.65 |
| 4 | 0.24 | 684.05 | 775.99 | 4.04 | 6.48 | 0.62 | 17.52 | 12.42 | 5.10 | 0.41 | 46.43 | 32.59 |
| 5 | 0.12 | 247.38 | 332.57 | 5.12 | 7.48 | 0.69 | 24.50 | 16.62 | 7.88 | 0.47 | 56.57 | 37.49 |
| 6 | 0.06 | 111.56 | 115.64 | 2.80 | 4.91 | 0.57 | 46.43 | 28.12 | 18.31 | 0.65 | 178.34 | 50.05 |
| 7 | 0.07 | 183.61 | 187.39 | 2.69 | 4.44 | 0.61 | 58.20 | 20.92 | 37.27 | 1.78 | 137.43 | 40.44 |
| 8 | 0.26 | 14.42 | 324.32 | 0.40 | 1.29 | 0.31 | 3.31 | 2.20 | 1.11 | 0.50 | 17.16 | 12.11 |
| 9 | 0.36 | 2633.79 | 9262.76 | 6.54 | 9.23 | 0.71 | 23.99 | 15.62 | 8.36 | 0.54 | 47.79 | 34.51 |
| 10 | 0.17 | 134.49 | 204.90 | 3.32 | 5.21 | 0.64 | 17.64 | 13.10 | 4.54 | 0.35 | 46.53 | 29.51 |
| 11 | 0.30 | 2662.15 | 3963.27 | 2.22 | 3.14 | 0.71 | 9.45 | 5.22 | 4.23 | 0.81 | 20.38 | 12.74 |
| 12 | 0.17 | 23.29 | 36.68 | 2.71 | 4.64 | 0.58 | 12.22 | 8.98 | 3.24 | 0.36 | 36.91 | 26.50 |
| 13 | 0.16 | 335.54 | 364.46 | 2.80 | 5.29 | 0.53 | 15.73 | 12.14 | 3.59 | 0.30 | 43.59 | 29.98 |
| 14 | 0.13 | 19.25 | 19.25 | 0.44 | 2.25 | 0.20 | 3.97 | 1.85 | 2.11 | 1.14 | 35.04 | 21.47 |
| 15 | 0.09 | 250.28 | 259.05 | 3.58 | 6.12 | 0.58 | 17.20 | 13.66 | 3.54 | 0.26 | 58.00 | 43.56 |
| 16 | 0.38 | 2706.80 | 3871.62 | 3.00 | 4.87 | 0.62 | 11.91 | 7.76 | 4.14 | 0.53 | 36.65 | 26.88 |
| 17 | 0.26 | 36.69 | 50.29 | 1.49 | 2.85 | 0.52 | 7.71 | 5.79 | 1.91 | 0.33 | 29.82 | 16.42 |
| 18 | 0.25 | 832.24 | 1440.64 | 2.42 | 4.27 | 0.57 | 13.70 | 9.98 | 3.72 | 0.37 | 59.04 | 44.97 |
| 19 | 0.14 | 39.82 | 53.93 | 4.62 | 6.85 | 0.68 | 21.13 | 16.31 | 4.82 | 0.30 | 52.92 | 38.76 |
| 20 | 0.19 | 184.69 | 218.06 | 5.76 | 8.55 | 0.67 | 23.56 | 18.34 | 5.23 | 0.29 | 64.64 | 44.93 |
| 21 | 0.33 | 89.69 | 260.15 | 3.00 | 5.61 | 0.54 | 24.85 | 14.70 | 10.14 | 0.69 | 211.53 | 106.18 |
| 22 | 0.22 | 299.39 | 353.29 | 2.25 | 3.11 | 0.72 | 12.78 | 10.07 | 2.71 | 0.27 | 20.86 | 9.85 |
| 23 | 0.29 | 270.61 | 411.15 | 4.24 | 6.56 | 0.65 | 17.23 | 13.24 | 3.99 | 0.30 | 52.49 | 32.30 |
| 24 | 0.28 | 215.32 | 250.62 | 4.47 | 6.98 | 0.64 | 18.19 | 13.87 | 4.33 | 0.31 | 50.25 | 34.87 |
| 25 | 0.29 | 241.71 | 272.53 | 4.77 | 7.01 | 0.68 | 19.52 | 15.15 | 4.38 | 0.29 | 46.84 | 31.96 |

**Table 2.** RBP test results on day 28 (d-28) and the physicochemical characteristics of 25 samples (Part 2).

| ADP No. | d-28 L/g VS | pH | CV MJ/kg TS | C % | H | N | Co | Fe mg/L | Mo | Ni | Total-COD g O$_2$/L | sCOD |
|---|---|---|---|---|---|---|---|---|---|---|---|---|
| 1 | 0.13 | 8.30 | 16.38 | 39.20 | 4.48 | 5.61 | 0.85 | 130.53 | 0.29 | 1.20 | 42.77 | 6.67 |
| 2 | 0.18 | 7.41 | 14.89 | 37.44 | 4.57 | 5.53 | 0.08 | 1144.15 | 0.30 | 0.50 | 40.35 | 2.73 |
| 3 | 0.13 | 8.35 | 18.61 | 34.80 | 4.52 | 7.34 | 0.24 | 1020.82 | 0.43 | 1.75 | 88.64 | 21.56 |
| 4 | 0.24 | 8.17 | 17.53 | 40.13 | 4.89 | 6.61 | 0.28 | 2031.83 | 0.19 | 0.60 | 44.04 | 12.14 |
| 5 | 0.12 | 8.45 | 16.13 | 38.32 | 4.20 | 6.05 | 0.21 | 90.76 | 0.18 | 0.39 | 40.69 | 15.16 |
| 6 | 0.06 | 8.14 | 3.29 | 17.08 | 1.71 | 2.20 | 1.08 | 2326.91 | 0.94 | 7.74 | 57.29 | 11.19 |
| 7 | 0.07 | 8.16 | 4.41 | 12.51 | 1.59 | 2.25 | 0.98 | 1923.48 | 0.69 | 5.06 | 39.32 | 8.45 |
| 8 | 0.26 | 7.33 | 20.26 | 46.25 | 5.58 | 6.11 | 0.03 | 87.36 | 0.07 | 0.24 | 28.62 | 3.16 |
| 9 | 0.36 | 8.35 | 22.26 | 47.75 | 5.32 | 8.51 | 0.09 | 539.07 | 0.14 | 0.50 | 75.39 | 35.94 |
| 10 | 0.17 | 8.04 | 14.78 | 36.85 | 4.00 | 5.74 | 0.42 | 174.88 | 0.30 | 0.71 | 36.12 | 7.82 |
| 11 | 0.30 | 7.62 | 17.02 | 38.87 | 4.43 | 6.30 | 0.04 | 210.67 | 0.08 | 0.19 | 28.69 | 10.00 |
| 12 | 0.17 | 8.10 | 18.26 | 42.43 | 4.88 | 6.13 | 0.25 | 234.07 | 0.08 | 0.28 | 31.68 | 10.17 |
| 13 | 0.16 | 8.37 | 16.21 | 38.47 | 4.47 | 6.65 | 1.04 | 111.12 | 0.30 | 1.02 | 46.88 | 9.53 |
| 14 | 0.13 | 7.50 | 15.25 | 32.94 | 4.50 | 5.78 | 0.45 | 1166.46 | 1.46 | 1.67 | 19.63 | 4.50 |
| 15 | 0.09 | 8.90 | 19.86 | 44.13 | 5.28 | 5.22 | 0.09 | 151.82 | 0.25 | 1.08 | 65.63 | 11.87 |
| 16 | 0.38 | 7.92 | 19.22 | 43.49 | 5.62 | 6.54 | 0.09 | 64.01 | 0.18 | 0.26 | 39.18 | 18.84 |
| 17 | 0.26 | 7.92 | 13.35 | 30.67 | 3.82 | 5.67 | 0.19 | 348.02 | 0.28 | 0.56 | 26.51 | 6.39 |
| 18 | 0.25 | 8.16 | 17.76 | 42.16 | 4.76 | 4.34 | 9.06 | 227.30 | 5.68 | 30.37 | 55.74 | 17.00 |
| 19 | 0.14 | 8.42 | 17.90 | 44.56 | 4.67 | 6.87 | 1.42 | 293.85 | 0.50 | 1.33 | 58.58 | 14.74 |
| 20 | 0.19 | 8.15 | 16.41 | 38.55 | 4.55 | 6.40 | 1.21 | 1056.09 | 0.45 | 1.55 | 68.52 | 15.57 |
| 21 | 0.33 | 8.07 | 12.24 | 33.21 | 2.64 | 2.35 | 2.58 | 4249.89 | 1.17 | 13.79 | 82.92 | 12.59 |
| 22 | 0.22 | 8.32 | 10.74 | 24.93 | 2.88 | 5.27 | 0.89 | 36.94 | 0.26 | 0.82 | 17.76 | 9.52 |
| 23 | 0.29 | 8.54 | 13.84 | 32.93 | 4.06 | 5.59 | 0.18 | 649.78 | 0.16 | 0.46 | 54.33 | 16.54 |
| 24 | 0.28 | 8.54 | 16.82 | 38.59 | 4.82 | 6.65 | 0.38 | 679.62 | 0.16 | 0.58 | 67.39 | 15.86 |
| 25 | 0.29 | 8.67 | 16.50 | 36.82 | 4.71 | 5.90 | 0.37 | 797.93 | 0.15 | 0.53 | 61.56 | 16.83 |

### 2.4.2. Assessing Prediction Uncertainty Using a Gaussian Process Regressor (GPR)

To quantify the uncertainty in the calculated biogas yield using the kinetic model predicted by DT, a Gaussian process regressor (GPR) method was applied in this study. GPR is a kernel-based Bayesian tool to perform nonlinear regression. The process is specified by a mean function m(x) and a covariance function k (x, x′ | θ) which defines the covariances between the responses at any two input locations x and x′. θ represents the hyperparameters of the covariance function and their values are learned from the training data by maximising the log marginal likelihood. Once the hyperparameters are decided, the prediction for new input is performed by computing the marginal posterior distribution conditioning on the dimensions with known inputs [25].

The principle of GPR is to predict one timestep further each time by fitting one more datum provided by the biogas-yield kinetic model and the prediction uncertainty bands were also returned. The predictions of the GPR with zero mean and squared exponential kernel and the GPR with linear basis function and squared exponential kernel were compared.

## 3. Results and Discussion

### 3.1. Digestate Characterisation and RBP Test Results

The 28-day RBP test results for each set of RBP samples together with the physicochemical characteristics of each digestate are shown in Tables 1 and 2. Specific cumulative biogas yield data collected during the 28-day testing period were reported elsewhere [9] and were used in the model fitting and training process in this study.

### 3.2. Assessing Strength of Fit for Biogas Production Models

Within the 28 days of the RBP test, it was noticeable that biogas yields of some samples in this study had reached a plateau, whilst others still were showing an upward trend close to the end of the 28-day test. This is clearly due to the different concentrations of readily degradable materials in the digestate samples. To distinguish these two types of digestate samples, the 25 digestate samples were classified into two types based on the absolute average of the daily biogas production change percentages in the last four days: (1) Type I: less than 0.5%; (2) Type II: more than 0.5% (Table 3).

**Table 3.** The classification of two types of RBP test biogas-production time series according to the absolute average of the daily biogas production change percentages in the last four days.

| | ADP | 3 | 4 | 5 | 11 | 12 | 13 | 16 | 19 | 20 | 21 | | | | |
|---|---|---|---|---|---|---|---|---|---|---|---|---|---|---|---|
| Type I | Increase (%) | 0.38 | 0.04 | 0.26 | 0.27 | 0.36 | 0.31 | 0.18 | 0.36 | 0.4 | 0.50 | | | | |
| | ADP | 1 | 2 | 6 | 7 | 8 | 9 | 10 | 14 | 15 | 17 | 18 | 22 | 23 | 24 | 25 |
| Type II | Increase (%) | 1.07 | 1.13 | 1.23 | 1.56 | 0.70 | 0.58 | 0.58 | 0.94 | 0.95 | 1.13 | 1.00 | 0.92 | 0.77 | 0.86 | 0.80 |

Table 4 shows the $R^2$ values for the fits of the three empirical biogas production models to RBP test data. Overall, the modified Gomperz model achieved lower $R^2$ values across the majority of the samples, indicating poorer fitting performance. In contrast, the PP first-order model could describe the biogas production of almost all samples with an $R^2$ value between 97–99%. The FO model performed better at describing Type I samples, whereas the PP first-order model was more suitable for substrates mixed with materials with different reaction-rate constants, and therefore performed better with Type II samples. However, it is worth noting that there are usually multiple sets of optimal solutions of the estimates of the PP first-order model's parameters ($Y_m$, $P$, $k_1$ and $k_2$). This is because the nonlinear least-square error function of this model when fitting a particular set of biogas yield data is not always convex. Therefore, it was not chosen for the study of training DTs to predict the model parameters from the digestate physicochemical characteristics in the following section.

**Table 4.** $R^2$ values of the fits of three empirical models (FO, MP and PP) to the Type I and Type II RBP test biogas-production time series ($R^2$ values larger than 97% are in bold).

| | | **Fitting $R^2$ Values (%)** | | | | | | | | | | | | | |
|---|---|---|---|---|---|---|---|---|---|---|---|---|---|---|---|
| | **ADP** | **3** | **4** | **5** | **11** | **12** | **13** | **16** | **19** | **20** | **21** | | | | |
| Type I | FO | **98.5** | **98.4** | **97.9** | 86.8 | **99.0** | **98.2** | **99.5** | **97.7** | 96.4 | **97.2** | | | | |
| | MG | **97.6** | 94.0 | **96.6** | 80.7 | 96.1 | 91.8 | 96.0 | 94.9 | 88.3 | 61.6 | | | | |
| | PP | **98.5** | **98.7** | **97.9** | 94.6 | **99.1** | **99.5** | **99.6** | **97.8** | **98.7** | **97.6** | | | | |
| | | **Fitting $R^2$ Values (%)** | | | | | | | | | | | | | |
| | ADP | 1 | 2 | 6 | 7 | 8 | 9 | 10 | 14 | 15 | 17 | 18 | 22 | 23 | 24 | 25 |
| Type II | FO | **97.1** | **98.6** | 86.5 | 77.2 | 83.2 | **97.3** | **98.4** | **97.6** | **97.5** | **97.5** | 95.5 | 96.0 | **98.8** | **99.1** | **99.5** |
| | MG | 88.8 | 93.9 | 78.6 | 85.8 | 68.2 | **99.1** | 93.0 | 90.8 | 74.9 | 88.4 | 85.6 | 85.9 | 96.7 | **97.3** | **99.0** |
| | PP | **99.9** | **99.6** | 92.8 | 92.6 | **98.0** | **97.4** | **98.9** | **99.5** | **98.9** | **97.8** | **99.1** | **98.9** | **98.8** | **99.1** | **99.5** |

In addition, when fitting the PP first-order model in Matlab using the least-square algorithm, the initial value set for the parameter $P$ should avoid 0.5 and $k_1$ and $k_2$ should not be the same. Otherwise, the partial derivatives of the error function with respect to $k_1$ and $k_2$ are the same, which means the moving direction of these two dimensions are the same and the nonlinear search for the minimum value of the error function value will settle at a local minimum point.

The fitting of the AR (1) model after the initial 3-day rapid biogas production stage was comparable to the PP first-order model, with $R^2$ values of 99% for the majority of samples (Table 5), regardless of Type I or Type II data. Figure 2 shows the fits of three empirical models and the AR (1) model to the cumulative biogas yield in RBP tests.

**Table 5.** $R^2$ values of the fits of the AR (1) model to the Type I and Type II RBP test biogas-production time series from day 4 ($R^2$ values larger than 97% are in bold).

| | **Fitting $R^2$ Values (%)** | | | | | | | | | | | | | | | |
|---|---|---|---|---|---|---|---|---|---|---|---|---|---|---|---|---|
| Type I | **ADP** | **3** | **4** | **5** | **11** | **12** | **13** | **16** | **19** | **20** | **21** | | | | | |
| | AR (1) | **98.2** | **99.4** | **97.4** | **97.0** | **98.6** | **99.3** | **99.8** | **99.7** | **98.9** | 90.1 | | | | | |
| | **Fitting $R^2$ Values (%)** | | | | | | | | | | | | | | | |
| Type II | **ADP** | **1** | **2** | **6** | **7** | **8** | **9** | **10** | **14** | **15** | **17** | **18** | **22** | **23** | **24** | **25** |
| | AR (1) | **99.2** | **99.8** | 85.4 | 80.9 | 95.9 | **99.1** | **99.5** | **99.7** | **98.9** | **99.1** | **99.6** | **99.5** | **98.7** | **99.1** | **99.9** |

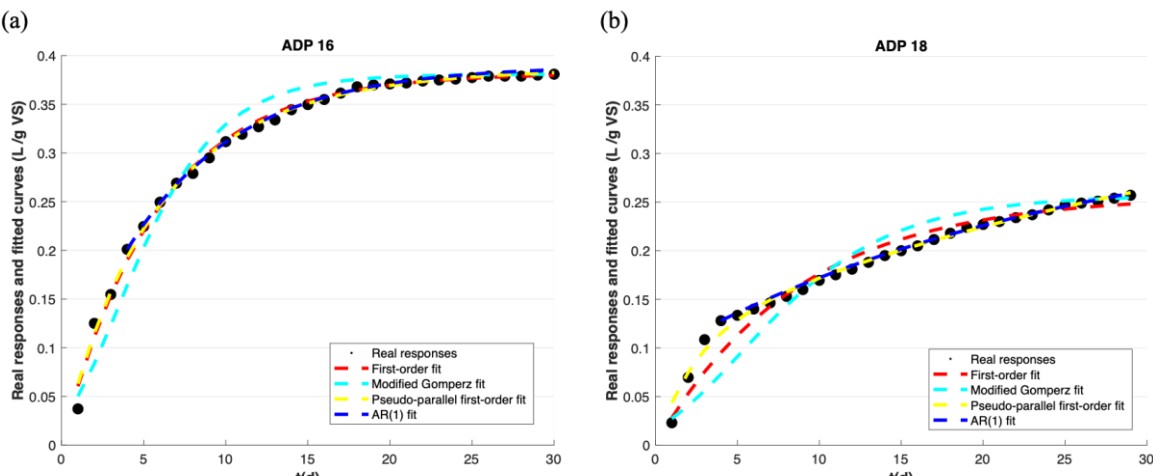

**Figure 2.** The fits of the three empirical models and the AR (1) model to the entire RBP test process for Type I (**a**) and Type II (**b**) data.

When using these four models to predict the final RBP results based on experimental data collected from a shorter test duration (5, 10, 15, 20 and 25 days), none of the three empirical models and the AR (1) model could achieve a sufficient level of accuracy to replace full-length RBP tests. This conclusion is in agreement with previous studies [12,26]. When using the first-order model, typically the level of prediction accuracy increases when more experimental data points are provided. FO performed moderately better than the MG model at predicting the RBP test result. This result was in agreement with the result of Nielfa et al. (2015) [26] for biomethane potential test biogas production prediction for the organic fraction of municipal solid waste and biological sludge co-digestion. However, both FO and MG models were proved to be unsuitable to predict the RBP result of Type II data.

Using 15 days of experimental data, the PP model could predict the RBP result with APE < 10% in nearly all the samples (both Type I and II). Therefore, the PP model is a preferred option for RBP test-result-prediction when the experiment is half-way through. For the AR (1) model, RBP data fitting and modelling starts from day 4, thus 10, 15, 20 and 25 days of experimental data were used for model fitting. The prediction ability of the AR (1) model was comparable to that of the PP model. Table 6 shows the predictions of RBP test results when fitting an increasing amount of experimental data to four biogas production models (due to the large sample numbers, only seven samples were randomly selected from each group of Type I and II data).

**Table 6.** The level of accuracy (expressed in APE%) of predicted 28-day RBP test result from fitting the mathematical models to an increasing number of experimental observations from 5 days to 25 days. APE less than 10% indicates accurate predication (in bold).

| Type I | Days | FO | MG | PP | AR (1) | Type II | Days | FO | MG | PP | AR (1) |
|---|---|---|---|---|---|---|---|---|---|---|---|
| ADP 3 | 5 | 77.16 | 24.23 | 35.08 | / | ADP 2 | 5 | 46.46 | 51.94 | 28.00 | / |
| | 10 | **4.98** | **9.62** | **7.26** | **1.15** | | 10 | 24.34 | 37.77 | 24.65 | 12.20 |
| | 15 | **1.01** | **5.14** | **2.70** | **0.47** | | 15 | 17.34 | 24.48 | **6.86** | **4.52** |
| | 20 | **2.10** | **1.55** | **0.88** | **1.71** | | 20 | 10.98 | 14.87 | **2.02** | **1.63** |
| | 25 | **2.29** | **0.07** | **3.23** | **1.81** | | 25 | **6.40** | **6.25** | **0.43** | **0.27** |
| ADP 4 | 5 | 57.99 | 26.65 | 52.67 | / | ADP 8 | 5 | **3.65** | 27.78 | **3.8080** | / |
| | 10 | **7.50** | 15.32 | **7.19** | **3.97** | | 10 | 15.24 | 17.56 | 15.13 | 12.57 |
| | 15 | **5.00** | **5.53** | **4.65** | **6.00** | | 15 | 12.03 | 12.02 | **8.31** | **8.62** |
| | 20 | **1.84** | **1.71** | **4.26** | **2.63** | | 20 | **9.62** | **9.62** | **5.23** | **8.20** |
| | 25 | **0.40** | **0.22** | **1.97** | **1.91** | | 25 | **3.48** | **3.44** | **2.76** | **5.74** |
| ADP 5 | 5 | 159.25 | 17.17 | 105.36 | / | ADP 10 | 5 | 163.39 | 37.62 | 29.58 | / |
| | 10 | **1.21** | 11.80 | **1.04** | **8.22** | | 10 | **7.63** | 25.74 | 19.18 | **2.99** |
| | 15 | **1.49** | **1.72** | **1.54** | **1.83** | | 15 | **7.73** | 16.41 | 12.75 | **0.03** |
| | 20 | **0.80** | **0.86** | **1.51** | **2.33** | | 20 | **5.40** | 10.20 | 21.58 | **0.83** |
| | 25 | **1.30** | **1.29** | **1.69** | **2.08** | | 25 | **3.49** | **6.22** | **4.85** | **1.02** |
| ADP 13 | 5 | 23.12 | 43.08 | 25.98 | / | ADP 14 | 5 | **0.16** | 44.20 | **5.68** | / |
| | 10 | 21.74 | 27.40 | **7.71** | **0.56** | | 10 | 27.04 | 32.57 | 27.06 | 44.65 |
| | 15 | 11.85 | 12.96 | 22.50 | **8.86** | | 15 | 17.04 | 19.33 | **1.20** | **8.78** |
| | 20 | **4.95** | **4.19** | 10.03 | **3.70** | | 20 | 10.67 | 10.33 | **5.62** | **3.70** |
| | 25 | **2.50** | **1.87** | **2.91** | **2.58** | | 25 | **6.58** | **4.85** | **1.92** | **2.04** |
| ADP 16 | 5 | 19.90 | 37.71 | 17.89 | / | ADP 17 | 5 | **1.6464** | 45.20 | 11.25 | / |
| | 10 | **4.66** | 18.19 | **4.76** | **5.69** | | 10 | 11.99 | 26.91 | **5.02** | 47.42 |
| | 15 | **3.38** | **8.20** | **3.05** | **0.57** | | 15 | 13.86 | 17.95 | **0.84** | **0.25** |
| | 20 | **1.09** | **2.65** | **3.70** | **1.37** | | 20 | 10.90 | 11.12 | **4.12** | **3.06** |
| | 25 | **0.61** | **0.95** | **1.00** | **0.89** | | 25 | **6.00** | 4.61 | **1.53** | **1.94** |
| ADP 19 | 5 | 183.89 | 24.27 | 146.92 | / | ADP 18 | 5 | **2.98** | 45.76 | **3.74** | / |
| | 10 | **2.33** | 13.07 | **2.33** | **8.62** | | 10 | 30.94 | 34.05 | 30.93 | **5.6969** |
| | 15 | **3.13** | **7.29** | **3.41** | **1.31** | | 15 | 22.36 | 22.19 | **7.51** | **0.57** |
| | 20 | **2.78** | **3.23** | **2.79** | **0.36** | | 20 | 12.64 | 11.80 | **4.16** | **1.37** |
| | 25 | **1.96** | **1.79** | **1.91** | **0.66** | | 25 | **6.29** | **4.44** | **2.96** | **0.89** |
| ADP 20 | 5 | 24.11 | 31.62 | 27.78 | | ADP 25 | 5 | 81.36 | 54.33 | 28.59 | / |
| | 10 | 11.16 | 21.66 | **8.64** | 48.84 | | 10 | 36.33 | 23.98 | **2.24** | 14.84 |
| | 15 | **8.93** | 13.72 | **4.54** | **0.62** | | 15 | **8.52** | 14.08 | **1.09** | **1.92** |
| | 20 | **7.15** | **9.66** | **0.06** | **1.15** | | 20 | **3.26** | **7.75** | **7.05** | **0.48** |
| | 25 | **5.32** | **6.69** | **1.12** | **0.51** | | 25 | **1.06** | **4.50** | **1.34** | **0.11** |

### 3.3. Biogas Yield Prediction from Digestate Physicochemical Characteristics by DT

DTs were first trained with the physicochemical characteristics of 20 digestate samples as predictors and the fitted coefficients and constants of AR (1) models as responses. A total of 5000 groups of splits between the training set and test set were randomly chosen to evaluate the prediction accuracy. The average MAPEs of the predicted AR (1) model coefficients and constants for among 5000 groups of test sets were 4.58% and 72.04%, respectively. The MAPE of the calculated RBP test result at day 28 from the predicted AR (1) model parameters was 52.125%.

With four more samples provided for the training set, the MAPEs of the predictions for the AR (1) model coefficient and constant with the LOO cross-validation method were 4.31% and 59.29%, respectively (Table 7). The MAPE of the calculated RBP test result at day 28 was 45.620%. With four more data provided, the prediction accuracy of the RBP test

result of the DTs improved 12.48%. Additionally, among 25 LOO cross-validation groups, the APEs of predicted RBP test results of five groups were smaller than 10%.

**Table 7.** The APEs for the AR (1) model coefficient and constant prediction and the biogas yield at 28th day prediction from the LOO cross-validation method.

| Test Sample No. | Fitted AR (1) Coefficient | Predicted AR (1) Coefficient | AR (1) Coefficient APE | Fitted AR (1) Constant | Predicted AR (1) Constant | AR (1) Constant APE | Real RBP Test Result | Predicted RBP Test Result | RBP Test Result APE |
|---|---|---|---|---|---|---|---|---|---|
| ADP1 | 0.932 | 0.937 | 0.5% | 0.010 | 0.010 | 1.1% | 0.132 | 0.137 | **4%** |
| ADP2 | 0.949 | 0.934 | 1.6% | 0.011 | 0.019 | 65.6% | 0.182 | 0.243 | 33.5% |
| ADP3 | 0.789 | 0.884 | 12% | 0.028 | 0.017 | 40.2% | 0.132 | 0.141 | **6.9%** |
| ADP4 | 0.876 | 0.896 | 2.2% | 0.030 | 0.035 | 17.5% | 0.236 | 0.321 | 36% |
| ADP5 | 0.834 | 0.850 | 1.9% | 0.020 | 0.036 | 84% | 0.117 | 0.236 | 101.9% |
| ADP6 | 0.946 | 0.813 | 14% | 0.004 | 0.011 | 174.7% | 0.062 | 0.057 | **7.9%** |
| ADP7 | 0.969 | 0.932 | 3.8% | 0.003 | 0.011 | 293% | 0.066 | 0.137 | 107.4% |
| ADP8 | 0.929 | 0.937 | 0.9% | 0.019 | 0.036 | 84.6% | 0.261 | 0.490 | 88% |
| ADP9 | 0.853 | 0.893 | 4.8% | 0.055 | 0.032 | 41.2% | 0.361 | 0.291 | 19.5% |
| ADP10 | 0.890 | 0.894 | 0.4% | 0.020 | 0.023 | 16.1% | 0.174 | 0.206 | 18.6% |
| ADP11 | 0.900 | 0.941 | 4.6% | 0.030 | 0.035 | 14.6% | 0.301 | 0.506 | 68.4% |
| ADP12 | 0.832 | 0.950 | 14.1% | 0.028 | 0.035 | 23.5% | 0.171 | 0.523 | 205.7% |
| ADP13 | 0.906 | 0.892 | 1.5% | 0.016 | 0.009 | 43% | 0.164 | 0.086 | 47.6% |
| ADP14 | 0.941 | 0.936 | 0.6% | 0.009 | 0.010 | 14.6% | 0.129 | 0.138 | **7.1%** |
| ADP15 | 0.901 | 0.847 | 6.1% | 0.009 | 0.020 | 123% | 0.086 | 0.126 | 46.5% |
| ADP16 | 0.864 | 0.842 | 2.5% | 0.053 | 0.032 | 38.7% | 0.379 | 0.205 | 46% |
| ADP17 | 0.934 | 0.936 | 0.2% | 0.019 | 0.009 | 52.8% | 0.257 | 0.139 | 45.9% |
| ADP18 | 0.961 | 0.870 | 9.4% | 0.013 | 0.041 | 211.6% | 0.254 | 0.307 | 21% |
| ADP19 | 0.861 | 0.898 | 4.2% | 0.020 | 0.023 | 16.5% | 0.139 | 0.212 | 52.5% |
| ADP20 | 0.887 | 0.894 | 0.8% | 0.022 | 0.022 | 1.8% | 0.194 | 0.202 | **4.3%** |
| ADP21 | 0.818 | 0.892 | 9.2% | 0.006 | 0.010 | 75.2% | 0.325 | 0.092 | 71.6% |
| ADP22 | 0.910 | 0.939 | 3.2% | 0.021 | 0.022 | 7% | 0.222 | 0.315 | 41.9% |
| ADP23 | 0.850 | 0.846 | 0.5% | 0.043 | 0.033 | 21.4% | 0.286 | 0.215 | 24.7% |
| ADP24 | 0.872 | 0.896 | 2.7% | 0.036 | 0.034 | 5.2% | 0.280 | 0.313 | 12% |
| ADP25 | 0.901 | 0.847 | 6.0% | 0.030 | 0.035 | 15.3% | 0.287 | 0.224 | 21.7% |
| MAPE | | | 4.3% | | | 59.3% | | | 45.6% |

*3.4. AR (1) Model Prediction Guide GPR*

When training the DTs with LOO cross-validation and ADP3 as the test set, the APE of the predicted biogas yield on day 28 using the inferred AR (1) model was 6.87%. This is used as an example for illustration in Figure 3. By accepting one more input each time from the inferred AR (1) model together with the experimental observations of the first three days, the predictions of every one timestep further of the GPR with zero mean and squared exponential kernel were smaller than AR (1)'s predictions for the first few timesteps and then gradually closer to the predictions of the AR (1) model. This was explained by the increased fitted GPR model length scale and vertical scale when receiving more training data. The smaller the length scale, the curvier the underlying function is, and the smaller the vertical scale, the more concentrated the underlying function is around the mean. In contrast, for the GPR with a linear basis function, the predictions surged for the first few timesteps and then approached the AR (1) model predictions. This corresponded to the decreased slope of the fitted linear mean function. In general, given that only one more timestep is predicted at a time, the selection of zero or linear mean function is not of much concern. The 95% confidence interval of the prediction of GPR narrowed when more data were provided.

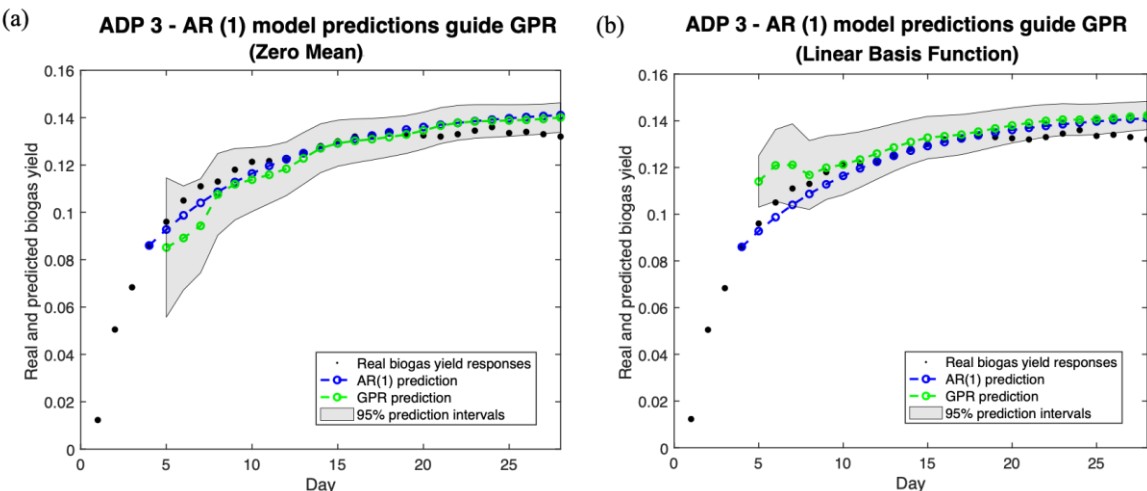

**Figure 3.** The prediction and its probability distribution of the GRP guided by the prediction of the inferred AR (1) model. (**a**) Zero mean and squared exponential kernel. (**b**) Linear basis function and squared exponential kernel.

## 4. Conclusions

This study has demonstrated that it is possible to use RBP experimental data collected in the initial stage of the test to predict the 28-day RBP result in a kinetic model fitting exercise. By fitting 15 days of experimental data from RBP tests to kinetic models, the PP first-order model and the AR (1) model achieved a promising accuracy with APE < 10%.

Further study demonstrated using the decision tree (DT) method that AR (1) model parameters can be predicted from the physicochemical characteristics of the digestate samples. This provides potential to further reduce the data requirement to four days of RBP experimental data and thereby significantly reduce the resting time of a standard 28-day RBP test to around four days. It was observed that when more training data were included in the DT machine learning model (from 20 to 24 samples), the prediction accuracy of the RBP result increased by 12.48%. This indicates that collecting more data to include in the model-training process can further improve the prediction outcome.

The framework of predicting kinetic model parameters from the physicochemical characteristics of the substrate can potentially be applied to the yield prediction of the product from other biochemical reaction processes.

**Author Contributions:** Conceptualization, W.G., P.L. and Y.J.; Methodology, Y.L., W.G. and Y.J.; Formal analysis, Y.L.; Writing—original draft, Y.L.; Writing—review & editing, W.G., P.L. and Y.J.; Supervision, W.G., P.L. and Y.J. All authors have read and agreed to the published version of the manuscript.

**Funding:** This research received no external funding.

**Institutional Review Board Statement:** Not applicable.

**Informed Consent Statement:** Not applicable.

**Data Availability Statement:** No new data were created or analysed in this study. Data sharing is not applicable to this article.

**Conflicts of Interest:** The authors declare no conflict of interest.

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
