# Peer review of "Shortening the Standard Testing Time for Residual Biogas Potential (RBP) Tests Using Biogas Yield Models and Substrate Physicochemical Characteristics"

_processes, doi:10.3390/pr11020441_

Round 1
Reviewer 1 Report
The paper has potential information for reducing the time for standard residual biogas potential tests. However, the paper can benefit from a grammar check or technical review. There are several concerns throughout the paper need to resolve.
Title
As per the study's title, "Shortening the Standard Residual Biogas Potential (RBP) Test Using Biogas Yield Models and Substrate Physicochemical Characteristics", - what is shortening? Therefore, the title of the manuscript might mislead the reader.
Abstract
Line13-14: Please rewrite the sentence to clear the statement.
Line14-16: The objective of the study needed to be clearer.
The concluding statement of the abstract needs to be included.
Introduction
Line 43: what is AD?
Line 58-60: Please provide the references for this statement.
Line 60: "None of the indicators" - A little speculative here.
Materials and Methods
Line 93: WW? Maybe MW.
Line 99: Please write TE fully at the beginning.
Line 100: IA/PA - please write a full name.
Line 108: TA - please write a full name.
Line 222: maximising?
Results and Discussion
Line 249-250: Incomplete sentence
Discussion is missing.
Limitations of the study can be added.
Conclusion
Line 357-359: Bad sentence
Please rewrite the full conclusion section to clear the message to the reader.
Reviewer 2 Report
Reviewer # comments:
Biogas digestate an abundant by-product of anaerobic digestion (AD) process, its management is of great significance. The manuscript investigated residual biogas potential of digestate adopting multivariate regression with decision trees (DTs) method to predict model parameters for the first-order autoregressive (AR (1)) model from substrate physicochemical parameters. The authors also used mean absolute percentage errors (MAPE) and Gaussian Process Regressor (GPR) to evaluate fitting the quality of the fitting model. The manuscript provided prediction model for shortening the standard RBP test and fits in the scope of the Processes Journal. However, the authors should consider the following comments for the improvement of the manuscript.
1. Page 2 line 79. What does “their large number” mean?
2. Page 2 line 81. There are many multivariate nonlinear regression analysis methods available. Why do you choose decision trees (DT)? What are its advantages? More details and references are needed.
3. Page 2 line 91. Since the physicochemical characteristics of digestate from different sources vary greatly, please give the distribution types of 25 sampling points. For example, biomass waste, food waste, livestock manure, co-digestion and so on. Make the article more valuable for reference.
4. Page 2 line 100. Because the whole text does not see the O element, please removed “O” from “elemental compositions (C, H, N, O)”.
5. Page 2 lines 99 and 102. Abbreviations, “TE, IA, PA”, that appear for the first time need to be given their full name.
6. Page 3 line 105. The procedure of VFAs determination and sample treatment should be given.
7. Page 3 lines 111 and 112. The name of the machine measuring TE concentrations should be given.
8. Page 3 line 150. “λ” is generally referred to as the lag time. “residence time” is easily mistaken for “hydraulic residence time” of AD.
9. Materials and Methods. This section should be supplemented with data processing methods and software used.
10. Table 1. Total VFA of ADP No.9, ADP No.11, ADP No.16 are 9262.76, 3963.27 and 3871.62 mg/L, respectively. In previous studies, it's reported that AD digestive system is suppressed, when total VFA is greater than 2000 mg/L. However, in this study, specific accumulative biogas yields of these three groups are higher than remaining groups. Please explain this phenomenon.
11. Table 2. pH value is from 14.42 to 2706.80. What do these values mean? How do you measure these values? In the column of C (%), these values are too small for total COD and TS of digestate. Please have a check.
12. Table 2. The unit of total COD and sCOD is kg O2/L? Besides, except for total COD and sCOD, the values of other columns are exactly the same as that of Table 1. Please check it carefully again.
13. Whether the data in Table 1 and Table 2 are from reference (WRAP, 2013)? Furthermore, since there may be many errors in Table 1 and Table 2, please check them carefully.
14. Table 2. The heading of Table 2 should be written at the top of the table.
15. Table 3, Table 4, Table 5, Table 6. None of these tables are complete. There's no way to evaluate them.
16. Page 3 line 250. “Modified Gomperz model performs the.”? Please complete this sentence.
17. Page 3 lines 265 and 266. A sentence should not be a paragraph of its own.
18. Page 3 lines 278~290. The source of the data was not identified. It should be explained in conjunction with the corresponding data table, so that readers can better understand the content.
19. Pages 13 and 14 lines 428~425. References, about WASTE & RESOURCES ACTION PROGRAMME (WRAP), should give the appropriate URL so that readers can get more detailed information.

Reviewer 3 Report
The manuscript titled " Shortening the Standard Residual Biogas Potential (RBP) Test Using Biogas Yield Models and Substrate Physicochemical Characteristics " has been sent to me for review. The article has good and valuable scientific content. The article is acceptable.
Author Response
No action requested from the reviewer.
Round 2
Reviewer 1 Report
Thank you for updating the manuscript. Only one point needs to address - please discuss how your results are better reduced the time, compared to published literature.
Author Response
We appreciate the reviewer’s positive comment.
The study has confirmed using PP model, RBP results can be predicted using 15 days of experimental data with a good level of accuracy (APE<10). This this a reduction of 13 days from a standard 28-day RBP test. This has been discussed in the paper (line 316-323).
In addition, the paper has proposed for the first time using DT method and feedstock material characterisation result to predict AR(1) model parameters. The finding of this work has demonstrated that it is feasible to reduce RBP experiment to 3 days using this modelling approach. This has been discussed in the paper (line 343-373)
Reviewer 2 Report
In the revised manuscript, the authors gave a good answer to the questions raised by the reviewer. This paper could be accepted.
Author Response
We appreciate the reviewer’s positive comment. No action required.